# A Pilot Feasibility Study of Reconnecting to Internal Sensations and Experiences (RISE), a Mindfulness-Informed Intervention to Reduce Interoceptive Dysfunction and Suicidal Ideation, among University Students in India

**DOI:** 10.3390/brainsci12020237

**Published:** 2022-02-09

**Authors:** April R. Smith, Shruti Kinkel-Ram, William Grunwald, Tony Sam George, Vaishali Raval

**Affiliations:** 1Department of Psychological Sciences, Auburn University, Auburn, AL 36849, USA; wrg0011@auburn.edu; 2Department of Psychology, Miami University, Oxford, OH 45056, USA; shankas@miamioh.edu (S.K.-R.); ravalvv@miamioh.edu (V.R.); 3Department of Psychology, Christ University, Bengaluru 560029, India; tony.sam.george@christuniversity.in

**Keywords:** interoception, interoceptive dysfunction, suicide, online intervention, India

## Abstract

Although 20% of the world’s suicides occur in India, suicide prevention efforts in India are lagging (Vijayakumar et al., 2021). Identification of risk factors for suicide in India, as well as the development of accessible interventions to treat these risk factors, could help reduce suicide in India. Interoceptive dysfunction—or an inability to recognize internal sensations in the body—has emerged as a robust correlate of suicidality among studies conducted in the United States. Additionally, a mindfulness-informed intervention designed to reduce interoceptive dysfunction, and thereby suicidality, has yielded promising initial effects in pilot testing (Smith et al., 2021). The current studies sought to replicate these findings in an Indian context. Study 1 (*n* = 276) found that specific aspects of interoceptive dysfunction were related to current, past, and future likelihood of suicidal ideation. Study 2 (*n* = 40) was a small, uncontrolled pre-post online pilot of the intervention, Reconnecting to Internal Sensations and Experiences (RISE). The intervention was rated as highly acceptable and demonstrated good retention. Additionally, the intervention was associated with improvements in certain aspects of interoceptive dysfunction and reductions in suicidal ideation and eating pathology. These preliminary results suggest further testing of the intervention among Indian samples is warranted.

## 1. Introduction

Over 800,000 people worldwide die by suicide each year. In India, which has the second largest population in the world, it is estimated that 170,000 people will die by suicide each year [1] This means over 20% of the world’s suicides occur in India. However, these rates may be considerable underestimates, as many suicides in India go unreported [2]. Given the high economic and psychological costs associated with suicide, accessible suicide treatments are needed. Mindfulness-based treatments have proven effective in reducing suicidality (e.g., Dialectical Behavior Therapy [DBT]; [3]). In part, this may be because mindfulness-based treatments improve interoception, or the awareness of internal sensations of the body, as poor interoception is robustly associated with suicidality [4,5,6]. However, many mindfulness-based treatments are costly to deliver in terms of time and resources; thus, more accessible mindfulness-based interventions are needed, particularly in low-to-middle income countries (LMIC) like India, where many individuals need treatment, but there are far too few trained professionals able to deliver it [7]. Thus, the purpose of the current studies was to first confirm whether poor interoception relates to suicidality among young adults in India, as has been repeatedly found in United States (U.S.) samples (Study 1). The second aim was to evaluate a mindfulness-informed intervention aimed at improving interoception and thereby reducing suicidality in an Indian context (Study 2).

Suicides in India have some notable differences from suicides in the U.S. For instance, unlike in the U.S., where suicides are most elevated among older adults (ages 45–64; [8]), suicides in India are particularly elevated among young adults (under the age of 30 [2,9,10]). Additionally, approximately equal numbers of men and women die by suicide in India, which differs from Western countries, where men are three times more likely to die by suicide than women [11]. Methods of suicide in India also differ from the U.S. Specifically, in India, hanging, ingestion of poison, overdose, and self-immolation are the top methods of suicide [2,12], but guns are the top method in the U.S. [13]. Given these differences, it is important to determine whether correlates of suicidality found in samples in the U.S., like interoceptive dysfunction, are also associated with suicidality among Indian samples. 

Interoception refers to awareness of internal sensations, like hunger, fullness, emotions, pain, and temperature [14]. There are individual differences in how well people can recognize interoceptive sensations. Some people with interoceptive dysfunction appear to be literally “out of touch” with their bodies. Research supports a role for interoceptive dysfunction in suicidality, with the idea being that individuals who are out of touch with their bodies are more able to engage in self-injurious behaviors, as it is much easier to harm something one is unattached to versus something one cares for [5,15,16]. The relationship between interoceptive dysfunction and suicidality has been replicated in several different types of samples from the U.S., including: two independent eating disorder samples, a military sample, a clinical outpatient sample, a sample of individuals engaging in non-suicidal self-injury, a sample of Black Americans, and an adolescent sample [4,5,6,17,18,19,20,21]. Further, interoceptive dysfunction has been found to vary as a function of suicide attempt recency, with the most recent attempters having the greatest impairment [4,5]. This finding suggests that interoceptive impairment may be an important factor to consider in the identification of individuals capable of engaging in lethal self-injury. However, to date, all studies on interoceptive dysfunction and suicidality have been conducted in the U.S.; thus, there is a need to determine whether interoceptive dysfunction is associated with suicidality among non-Western samples. Given that suicides in India are elevated, and particularly so among young adults, it is important to test this association among young adults in India. Thus, the first aim of the current set of studies was to test whether interoceptive dysfunction associates with suicidality among university students in India.

If interoceptive dysfunction remains associated with suicidality among young adults in India, then this would suggest a role for interoceptive dysfunction in treating suicidality. In terms of treatments for suicidality, mindfulness-based interventions hold great promise. In fact, a promising treatment for suicidal behavior—DBT [3]—includes mindfulness as the foundation for all other skills. One aim of mindfulness is to increase awareness of internal sensations. As noted, it is possible that DBT reduces suicidal behavior in part because it improves interoception. Support for this comes from a study of active-duty U.S. military personnel, which found that an eight-week mindfulness training program affected brain structures associated with enhanced interoceptive processing [22]. Although DBT is effective at reducing suicidal behavior, implementing DBT takes considerable resources and time. Evidence-based treatments for suicide that are brief, affordable, and portable have the ability to reach more people, and may thus be well-suited for the Indian context.

Given the need for accessible interventions for suicide, we developed a novel digital intervention to improve interoceptive dysfunction, called Reconnecting to Internal Sensations and Experiences (RISE). The RISE intervention was specifically designed to be accessible and easy to implement, even with few resources. In fact, all that is needed for the training is access to an Internet-enabled device (e.g., smartphone, laptop, tablet). Given that close to half of India’s population has Internet access, and this percentage is steadily rising, this makes RISE an attractive option in this setting [23,24]. Because our intervention is designed to be portable, freely accessible, and brief, it can also overcome barriers to traditional face-to-face treatment. This is critically important in India, given that many are unable to access treatment, often due to issues like cost, accessibility, and resources [7,25,26].

The RISE intervention was specifically designed to improve interoception, and in so doing, reduce suicidality. Notably, we conducted a pilot trial of RISE, which demonstrated its acceptability and feasibility among clinical outpatients in the U.S. [27]. This pilot was a small (*n* = 22), uncontrolled pre-post pilot of RISE. Within this U.S. clinical sample, RISE was associated with significant improvements in interoception and reductions in suicidal ideation, as well as reductions in general psychological symptoms. However, it remains to be determined whether RISE will be associated with similar improvements in other samples, including young adults in India. Thus, the second aim of the current set of studies was to conduct a pilot test of RISE among young adults in India. 

## 2. Study 1

In the first study, we sought to first replicate findings from U.S. samples demonstrating links between interoceptive deficits and suicidality. We predicted that greater interoceptive dysfunction would relate to greater suicidality.

### 2.1. Methods

#### 2.1.1. Participants and Procedure

An a priori power analysis in G*Power [28] indicated that we would need at least 191 participants to detect a small effect (d = 0.2) at 80% power. We over-sampled in order to allow for participant attrition due to dropping out or failure to pass attention checks. Thus, participants (*n* = 465) were Indian adults who were recruited via online advertisements and local advertising at Christ University in Bangaluru, Karnataka in India. Participants were included if they were above 18 years of age, were fluent in English, of Indian origin, and living in India. There were no additional exclusionary criteria. This study was approved by the Miami University Institutional Review Board (#01656r) and informed consent was obtained online at the beginning of the study from all participants. Participants were asked to complete a battery of self-report assessments on Qualtrics. The average time taken to complete the survey was 56 min. Participants who consented to participate were provided Amazon India Gift cards worth 450 Indian Rupees as compensation for their participation in the study, which is equivalent to approximately $6.50. 

To ensure data quality, three attention checks were placed throughout the online survey (e.g., *If you are reading attentively, select option seven*). Participants who failed two or more attention checks were screened out of the study, leaving us with a final analytic sample of 276 individuals. Participants were 52.9% male and ranged from 18 to 35 years of age (*M_age_ =* 20.62; *SD* = 2.22). 

#### 2.1.2. Measures

Multidimensional Assessment of Interoceptive Awareness-2. The Multidimensional Assessment of Interoceptive Awareness (MAIA-2) [29] was used to measure interoception. Participants are asked to self-report their own awareness of sensations and signals originating within their own bodies (e.g., *I listen for information from my body about my emotional state*). The MAIA-2 is a 37-item state-trait assessment of interoception across 8 domains: Noticing, Not-Distracting, Not-Worrying, Attention Regulation, Emotional Awareness, Self-Regulation, Body Listening, and Body Trust. Responses can range from 0 (*Never*) to 5 (*Always*), and higher ratings are indicative of greater awareness of bodily signals and sensations; thus, *lower* scores indicate more interoceptive dysfunction. In the current study, the internal consistency of the overall measure was good (α = 0.87). The internal reliabilities of the subscales were as follows: Noticing (0.76), Not-Distracting (0.87), Not-Worrying (0.56), Attention Regulation (0.86), Emotional Awareness (0.85), Self-Regulation (0.85), Body Listening (0.85), and Body Trust (0.87). Notably, the low internal consistency of the Not-Worrying subscale is in keeping with other studies that have found similarly low reliabilities for this subscale [29,30]. 

Depressive Symptom Inventory-Suicidality Subscale. The Depressive Symptom Inventory-Suicidality Subscale (DSI-SS) [31] was used to assess current suicidal ideation. The DSI-SS is a self-report assessment of suicidal thoughts and urges. It includes four items which are rated on a four-point Likert scale, with higher scores indicative of greater severity of suicidal ideation. Participants are asked to select the statement that best fits how they have felt in the past two weeks. For example, item 1 ranges from 0 (*I do not have thoughts of killing myself*) to 3 (*I always have thoughts of killing myself*). This measure has demonstrated good internal consistency in young adult samples from India in prior research (α = 0.87) [32], and internal consistency in the current study was excellent (α = 0.91).

Self-Injurious Thoughts and Behaviors Interview-Revised. Items from the Self-Injurious Thoughts and Behaviors Interview-Revised (SITBI-R) [33] were used to assess for lifetime suicidal ideation, future suicidal ideation likelihood, and lifetime suicide attempts. The SITBI is an assessment with a structured interview and self-report format measuring the extent to which an individual endorses various suicide-related thoughts and behaviors. All 115 items were administered to participants, but only the three items related to lifetime suicidal ideation, future suicidal ideation likelihood, and lifetime suicide attempts were used in the current study (e.g., *What is the likelihood that you will have thoughts of killing yourself in the future?*). Lifetime suicide attempts and lifetime suicidal ideation contained binary (*yes-no)* response options, while the item on future suicidal ideation likelihood ranged from *Low/Little (0)* to *Very Much/Severe (4).*

#### 2.1.3. Data Analytic Plan

There were almost no missing data for variables of interest (i.e., only one participant was missing data for the Body Trust subscale of the MAIA), thus, missing data were handled via listwise deletion. MAIA subscales exhibited acceptable skew and kurtosis. However, as is to be expected in a non-clinical sample, suicidal ideation scores were skewed and kurtotic, indicating many people endorsed zero suicidality; thus, scores on our suicidal ideation measure (DSI-SS) were square-root transformed. However, the results between the transformed and non-transformed DSI-SS variable did not appreciably vary; thus, the non-transformed results are reported for ease of interpretation. All analyses were performed using SPSS version 27 [34] and Cohen’s D effect sizes are reported.

In order to test our hypothesis that interoceptive dysfunction would relate to suicidality, we computed Pearson correlations between MAIA subscales and suicidal ideation (as assessed by the DSI-SS), see Table 1. Spearman correlations were used to assess the relation between MAIA subscales and presence or absence of past ideation. We also computed correlations between MAIA subscales and future likelihood of having thoughts about suicide, as assessed by the SITBI. Notably, for this analysis, only participants who endorsed ever having thoughts of suicide (*n* = 98) rated their future likelihood of ideation. Further, as there were few individuals with a history of suicide attempt (*n* = 23), we compared interoception scores between those with any reported current or lifetime history of suicidal ideation or attempt (i.e., Suicidality Group, *n* = 102) and those with no reported history of suicidal ideation or attempt (i.e., No Suicidality Group, *n* = 174). We then calculated *t*-tests and effect sizes to test whether the Suicidality Group had worse interoceptive dysfunction than the No Suicidality Group.

### 2.2. Results and Discussion

Partially confirming our first hypothesis, MAIA Not-Distracting (*r* = −0.18, *p* = 0.003), MAIA Not-Worrying (*r* = −0.12, *p* = 0.045), and MAIA Body Trust (*r* = −0.22, *p* < 0.001) were significantly associated with current suicidal ideation as assessed by the DSI-SS; see Table 1. In other words, the more individuals distracted from sensations of pain or discomfort, the more individuals worried about or experienced emotional distress regarding sensations of pain or discomfort, and the less individuals trusted their bodies, the more suicidal ideation they experienced. We also found that MAIA Not-Distracting (r = −0.26, *p* = 0.009) and MAIA Body Trust (r = −0.38, *p* < 0.001) were related to likelihood of experiencing future suicidal ideation. 

Also in partial support of predictions, those in the Suicidality group had lower scores on Not-Distracting, *t* (274) = 2.59, *p* = 0.001, *d* = 0.32 and Body Trust, *t* (274) = 3.74, *p* < 0.001, *d* = 0.47, relative to the No Suicidality Group. 

Overall, our predictions were partially confirmed. Although not all aspects of interoception as assessed by the MAIA correlated with suicidal ideation, specific facets did: distracting from and worrying about sensations of pain and discomfort, as well as lacking trust in one’s body. The size of these associations ranged in magnitude from small to medium, though most were small. Further, we found that those with a history of either suicidal ideation and/or attempt had worse body trust and more of a tendency to distract from sensations of pain and discomfort than those without a suicidality history. These findings partially replicate Rogers and colleagues’ [35] findings, as all three of these facets were also found to be diminished in individuals with suicidality histories relative to those without in their study. However, Rogers and colleagues [35] also found that self-regulation was significantly lower among those with suicide histories relative to those without, whereas we did not find that difference in our sample. 

These findings should be considered in light of important limitations. For instance, the cross-sectional nature of the study precludes drawing any conclusions about directionality. Further, interoception was self-reported, which may be particularly problematic for individuals with poor interoception. Suicidality was also self-reported; however, we did implement a well-validated assessment of suicidality in order to improve upon single-item assessments of suicide history [33,36,37]). Moreover, a large number of participants were excluded from analyses based on their performance on attention checks. Additionally, it will be important for future work to assess relations between interoception and suicidal behaviors among individuals in India, given that suicidal behaviors are more predictive of future self-injury than ideation [38].

## 3. Study 2

Study 1 demonstrated that aspects of interoceptive dysfunction were related to current, lifetime, and future likelihood of suicidal ideation among young adults in India. Given these results, a treatment like RISE, that targets interoceptive dysfunction via mindfulness-based exercises, holds promise in reducing suicidality among young adults in India. In fact, various mindfulness-based interventions have been found to improve interoception and both psychological and physical conditions [39,40,41].

Based on theory and experimental work, RISE targets several aspects of interoception, including bodily awareness, body functionality, emotion awareness, and intuitive eating. Specifically, the training starts out by introducing the concept of interoception and reviewing how it relates to mental and physical health. Participants then learn about body awareness by engaging in a six-minute guided progressive muscle relaxation (PMR) exercise. The first session also introduces the concept of “body functionality” or considering the body for what it can do, rather than what it looks like [42,43]. Participants then engage in interactive reading and writing exercises about body functionality related to physical activity and movement as well as creative endeavors. Each subsequent session begins with the PMR exercise and reviews the material from the previous week. In the second session, participants learn about the function of emotions and review associated biological changes and experiences associated with several different emotions. They further review how to shift an emotional experience if it is associated with ineffective behavioral actions by focusing on their breathing and using positive emotional “touchstones” [44]. In the third session, participants return to the concept of body functionality as it relates to self-care and communication. The fourth and final session introduces and reviews the concepts and practices of intuitive eating, or using cues from the body to guide eating, and incorporates a mindful eating exercise. The fourth session ends by reviewing all the material covered and having the participant consider ways to incorporate the skills and knowledge gained in their own life. 

Given that interoception is related to suicidality among young adults in India, and that RISE was associated with clinical improvements among a U.S. sample of clinical outpatients [27], an appropriate next step is to test RISE in an Indian sample. Informed by Purpose Guided Trial Design [45] and the phase of development of the novel RISE intervention, our primary aim was to evaluate acceptability and feasibility of the RISE intervention in this novel sample. Thus, we chose to evaluate RISE via an uncontrolled pre-post design among young adults in India. In addition to examining its acceptability and feasibility, we also tested whether interoception and suicidal ideation improved from pre-intervention to post-intervention. As an exploratory aim, we also assessed whether other psychological symptoms improved. We predicted that participants would find RISE to be acceptable, and that we would see improvements in interoception and reductions in suicidal ideation.

### 3.1. Method

#### 3.1.1. Participants and Procedure

A subset of the original sample from study 1 (*n* = 50) was randomly selected and invited to participate in a longitudinal study to test the acceptability and feasibility of the RISE intervention. Of those invited, 40 participants enrolled. Hence, recruitment information and inclusion and exclusion criteria were identical to study 1. Participants completed the four intervention modules of RISE over the course of four weeks (one module per week). Thirty-one participants completed all four intervention modules; these participants also passed the attention checks, and hence were used in the analyses for this current study. The final sample was predominantly male (80.6%) and ranged from 18 to 29 years of age (*M_age_ =* 21.03; *SD* = 2.37).

Participants completed baseline and post-study questionnaires at the beginning and end of the study respectively. Study procedures were approved by the Miami University Institutional Review Board (#01656r). All study procedures were conducted online and participants were emailed by the study team with reminders to complete the RISE interventions. The RISE trainings were delivered via audio and written exercises that were embedded into the Qualtrics surveys that were e-mailed to participants. Participants received 900 Indian Rupees (INR; approximately $13) for completing modules 1 and 4 and INR 450 for completing modules 2 and 3 (approximately $6.50). Participants also received an INR 900 bonus payment for completing all four time points, and INR 450 bonus for completing three time points. All payments were made via Amazon India gift cards. 

#### 3.1.2. Measures

##### Primary Outcomes

*Treatment Acceptability Questionnaire (TAQ;)* [46]. Participants were asked to answer three adapted questions from the TAQ to assess the acceptability of RISE. Questions were rated on a 7-point Likert scale. One question assessed acceptability, “*Overall, how acceptable did you find the modules that you completed over the last few weeks to be?*”, another effectiveness, “*How effective do you think these modules might be?*”, and a third potential negative side effects, “*How likely do you think it is that the modules may have had negative side effects?*”. Responses to these quantitative questions were used in the analyses to examine the feasibility and acceptability of RISE. 

##### Secondary Outcomes

*MAIA.* See Study 1 for a description of the MAIA and DSI item content. In the current sample, the internal consistency of the overall measure was excellent at post-intervention (α = 0.94). The internal reliabilities of the MAIA subscales ranged at post-test as follows: Noticing (0.84), Not-Distracting (0.79), Not-Worrying (0.65), Attention Regulation (0.92), Emotional Awareness (0.79), Self-Regulation (0.86), Body Listening (0.75), and Body Trust (0.89).

*DSI.* In the current sample, the internal consistency of the overall measure was excellent when completed at post-intervention (α = 0.87).

*The Brief Symptom Inventory (BSI)* [47]. The BSI was used to measure psychological distress. Participants report to what extent they endorse a variety of distress-related symptoms (e.g., *Feeling tense or keyed up*). Responses can range on a 5-point Likert scale from 0 (*Not at all*) to 4 (*Extremely*), and higher ratings are indicative of greater psychological distress. In the current study, the internal consistency of the overall measure was excellent at post-intervention (α = 0.94). 

*Eating Pathology Symptom Inventory.* The Eating Pathology Symptom Inventory (EPSI) [48] was used to measure disordered eating symptoms, such as restricting food and purging. The EPSI is a 45-item assessment of eating pathology across eight domains: Body Dissatisfaction, Binge Eating, Cognitive Restraint, Excessive Exercise, Restricting, Purging, Muscle Building, and Negative Attitudes Toward Obesity. Participants are asked to self-report on the extent to which they endorse different eating disorder-related thoughts and behaviors over the past four weeks (e.g., *I made myself vomit in order to lose weight*). Responses can range from 0 (*Never*) to 4 (*Very Often*), and higher ratings are indicative of greater severity of disordered eating symptoms. In the current study, the internal consistency of the overall measure was excellent at post-intervention (α = 0.95). 

#### 3.1.3. Data Analysis

To examine the feasibility and acceptability of RISE, mean scores on the items from the TAQ were examined. In order to examine the effects of RISE, as with the initial Smith et al. (2021) pilot of RISE, we conducted a per-protocol analysis. As such, we examined mean differences from pre-test to post-test among the participants who completed both pre- and post-treatment measures (*n* = 31). Specifically, paired sample *t*-tests were conducted to examine within-person changes on our primary and secondary clinical outcomes. 

#### 3.1.4. Results and Discussion

##### Intervention Feasibility

The intervention was considered feasible if more than 75% of participants that initiated the intervention completed it. The consent rate was 75% (40/50) and the participation rate was 78% (31/40). Forty participants completed the pre-test assessment and session 1; 31 individuals completed session 4 and the post-test assessment, thus the attrition rate was 22.5%. There were few demographic or clinical differences between those who completed vs did not complete session 4. However, those who completed the intervention were more likely to be male and had lower scores on the MAIA Not Distracting subscale than those who did not complete the intervention. 

##### Intervention Acceptability

The RISE treatment demonstrated excellent acceptability, *M* = 6.35, *SD* = 0.76. Additionally, the treatment was rated as effective (*M* = 5.84, *SD* = 1.07) and unlikely to produce negative side effects (*M* = 1.68, *SD* = 1.01). 

##### Secondary Clinical Outcomes

In line with hypotheses, all aspects of interoceptive dysfunction as assessed by the MAIA improved from pre to post-test; however, only some of these reached statistical significance. Specifically, there were significant improvements in Body Listening (*d* = −0.43) and Noticing (*d* = −0.39). Not-Distracting (*d* = −0.30) also improved at a trend level. See Table 2 for a summary of results.

Mean suicidal ideation scores also decreased at a trend level; the effect size was small to medium (*d* = 0.31). 

##### Additional Clinical Outcomes

Total eating disorder symptoms also improved significantly, (*d* = 0.48). Mean BSI scores decreased, but not to a significant degree, *d* = 0.16.

Overall, our results provide evidence for the feasibility and acceptability of the RISE intervention among a sample of young adults in India. The RISE intervention was rated as highly acceptable; it was also rated as likely to be effective and unlikely to have any negative side effects. Further, RISE was descriptively rated as even more acceptable and effective, and less likely to have had iatrogenic effects, than was initially reported in Smith and colleagues [27]. Additionally, RISE was associated with significant improvements in several aspects of interoception. Further, RISE was associated with significant improvements in eating disorder symptoms. However, there are some notable limitations that must be kept in mind when interpreting results; these limitations also provide directions for future research. Primarily, findings are limited by the relatively small sample size, uncontrolled nature of the study, and lack of a longer-term follow-up.

## 4. General Discussion

The aims of these two studies were to investigate the relationship between interoceptive dysfunction and suicidality, as well as a novel mindfulness-informed intervention for improving interoception, among young adults in India. Largely replicating findings from the U.S., we found that certain aspects of interoceptive dysfunction were related to suicidal ideation. Further, we found that the RISE intervention was rated as highly acceptable and had good retention across the four weeks. RISE was also associated with improvements in certain aspects of interoception, as well as improvements in suicidal ideation and eating disorders symptoms. 

In our first study, we found that particular components of interoception related not only to current suicidal ideation, but also to lifetime ideation and belief in the likelihood that one would experience suicidal ideation in the future. Specifically, not-worrying, or the ability to tolerate pain and discomfort, not-distracting, or the tendency to attend to sensations of pain or discomfort, and body trusting, or experiencing the body as safe and trustworthy, were negatively related to suicidal ideation. These findings largely replicate a prior study of associations between interoceptive dysfunction and suicidality, which also found negative relations between not-distracting, not-worrying, and trusting and suicidal ideation in a U.S. sample [35]. Rogers and colleagues [35] additionally found a negative relation between self-regulation and suicidal ideation, which was not found in the current study. Worrying about and distracting from sensations of pain and discomfort relate conceptually to low distress tolerance, which is related to suicidal ideation [49,50]. Additionally, as in the current study, several other studies have found negative relations between trusting and suicidal ideation [51,52]. In fact, a recent meta-analysis indicated that lack of trusting in one’s body appears to be the facet of self-reported interoceptive dysfunction most consistently linked with suicidality [53]. Overall, our findings with young adults from India supported prior research based on U.S. samples demonstrating an association between interoceptive dysfunction and suicidal ideation. 

In study 2, we tested whether RISE, an intervention designed to improve interoception, would be acceptable and feasible to deliver in an Indian context; our results were largely supportive. In fact, the modal response regarding RISE acceptability was a 7, the highest option available. Additionally, RISE was viewed as being effective and very unlikely to have negative effects. Further, overall retention in the study was good, as 31 out of 40 participants completed the intervention. This attrition rate of 22% is better than the attrition rate of 31% in the first U.S. pilot of RISE [27].

Our hypothesis that RISE would be associated with improvements in interoception and decreases in suicidal ideation was partially confirmed. All aspects of interoception improved from pre-intervention to post-intervention; however, only noticing and body listening did so to a significant degree, and not-distracting improved at a trend level. The size of these effects were in the small to medium range. Further, the MAIA subscales associated with suicidal ideation were not the same subscales that demonstrated the largest improvements after completing RISE in this sample. These somewhat mixed findings are consistent with other literature on mindfulness-based training and interoception, which has also produced mixed results, and suggest that effects may vary depending on the component of interoception under examination. For example, some studies have not found effects of meditation on cardiac interoceptive accuracy [54]. Thus, it will be important for future research with larger samples to test whether improvement in aspects of interoception lead to reductions in suicidal ideation. It is also worth noting that the first pilot of RISE, which was conducted among a small sample of clinical patients in U.S., also found that interoception improved; however, different aspects of interoception improved. Specifically, Smith and colleagues [27] found that RISE was associated with better emotional awareness and self-regulation. Future testing is needed in order to determine if these differences may be due to the use of a non-clinical sample and/or a non-U.S. sample. Further, Smith and colleagues [27] found that RISE was associated with significant decreases in suicidal ideation, though in the current study, ideation improved at a trend level with a small to medium effect size. The use of a non-clinical sample in the current study, with overall low levels of suicidal ideation, may have contributed to this effect failing to reach significance. 

Additionally, replicating Smith and colleagues [27], we found that RISE was associated with improvements in eating disorder symptoms; however, we did not find significant improvements in general psychological symptoms as were found in the previous pilot. Interoceptive dysfunction is robustly linked to eating disorders [55,56]. Further, recent research suggests that interoceptive dysfunction may be a core component of eating disorder pathology [57,58]. Thus, it is perhaps not surprising that the RISE intervention is associated with both decreases in suicidal ideation and eating disorder symptoms. If further testing continues to demonstrate improvements on these clinical outcomes, this could indicate that RISE would be an appropriate treatment for suicidal ideation and eating disorders, as well as co-occurring disordered eating and suicidality. Notably, there are high rates of suicidality among individuals with eating disorders [59,60,61,62], yet existing treatments for these conditions do not target this comorbidity. Therefore, it will be important for future research to determine whether RISE may be an appropriate treatment for co-occurring disordered eating and suicidality. 

### Strengths, Limitations and Future Directions

This was the first published study, to the best of our knowledge, to test an association between interoceptive dysfunction and suicidal ideation in a non-U.S. sample. In addition, we tested acceptability and feasibility of the RISE intervention, and replicated previous findings demonstrating that RISE is associated with improvements in some aspects of interoceptive dysfunction, suicidal ideation, and other clinical outcomes, such as disordered eating. Our second study testing the RISE intervention was limited by a small sample size and not having a long-term follow-up. Further, a majority of participants who completed the full intervention were male. Additionally, we employed a sample of university students, who may not be representative of young adults in India. Non-university students are more likely to live with family and may not have as much privacy to engage in online treatment sessions as compared to university students. Moreover, the psychometrics of the MAIA have not been examined in an Indian sample, and there may be differences in interoception that are culturally influenced [63]. However, apart from the Not-Worrying subscale, which typically demonstrates the lowest reliability in other Western studies as well, the reliabilities for the MAIA subscales were good to excellent. Despite these limitations, this design was purposeful, and provided preliminary evidence of acceptability, feasibility, and potential efficacy of RISE for Indian young adults. As next steps, it will be important to test the RISE intervention in larger samples of young adults in India where RISE can be compared directly to clinically meaningful alternatives such as treatment as usual, and where longer-term follow-up is included. Studies using larger samples from India will also be helpful to test change in interoception as a purported mechanism of action.

## 5. Conclusions

Aspects of interoceptive dysfunction are associated with suicidal ideation among young adults in India, suggesting that interventions aimed at improving interoceptive awareness can be beneficial in reducing mental health burden related to suicide. Preliminary evidence suggests that Indian young adults considered the RISE intervention to be acceptable and that it is feasible to implement. In addition, RISE improved some features of interoception and has the potential for reducing clinical outcomes in India. Future work is needed to replicate our findings with larger samples and longer follow-up.

## Figures and Tables

**Table 1 brainsci-12-00237-t001:** Zero-order correlations for all MAIA subscales and current, past, and future likelihood of suicidal ideation symptoms in Study 1.

	1	2	3	4	5	6	7	8	9	10	11
*Mean (SD)*	2.53 (1.19)	2.32 (1.18)	2.43 (0.89)	2.63 (0.97)	3.17 (1.16)	2.87 (1.20)	2.25 (1.26)	3.31 (1.24)	0.67 (1.59)		
1. Noticing	−										
2. Not Distracting	−0.41 **	-									
3. Not Worrying ^+^	−0.18 **	0.11 ^+^	-								
4. Attention Regulation	−0.43 **	0.21 **	0.16 **	-							
5. Emotion Awareness	0.52 **	−0.29**	−0.15 *	0.55 **	-						
6. Self-Regulation	0.36 **	−0.09	0.11 *	0.55 **	0.62 **	-					
7. Body Listening	0.46 **	−0.11	−0.03	0.53 **	0.58 **	0.62 **	-				
8. Body Trust	0.25 **	−0.05	0.12 *	0.51 **	0.44 **	0.55 **	0.44 **	-			
9. Current SI	0.10	−0.18 **	−0.12 *	−0.04	0.00	−0.08	−0.01	−0.22 **	-		
10. Past SI	0.06	−0.16 **	−0.08	−0.07	−0.03	−0.09	−0.02	−0.23 **	0.52 **	-	
11. Future SI ^++^	0.13	−0.26 **	−0.12	0.05	0.06	−0.13	−0.08	−0.38 **	0.66 **	X	-

*Note*. Items 1–8 refer to MAIA subscales; SI = Suicidal Ideation; ^+^*p* = 0.06, * *p* < 0.05, ** *p* < 0.01. ^++^ Future SI question was only answered by those who endorsed past SI (*n* = 98), as such a correlation between past and future SI cannot be computed.

**Table 2 brainsci-12-00237-t002:** Study 3 paired *t*-test results of Reconnecting to Internal Sensations and Experiences intervention among participants who completed the pre and post assessments (*n* = 31).

	Pre	Post			
	*M (SD)*	*M (SD)*	*t*	*p*	*D*
Interoception					
Attention Regulation	2.76 (1.01)	2.93 (1.11)	−1.02	0.31	−0.18
**Body Listening**	**2.41 (1.11)**	**2.90 (1.10)**	**−2.39**	**0.02**	**−0.43**
Emotional Awareness	3.34 (1.10)	3.54 (1.00)	−1.28	0.21	−0.23
Not Distracting	2.09 (1.06)	2.39 (1.03)	−1.64	0.11	−0.30
**Noticing**	**2.37 (1.19)**	**2.77 (1.12)**	**−2.17**	**0.04**	**−0.39**
Not Worry	2.59 (1.03)	2.74 (0.95)	−1.04	0.31	−0.19
Self-Regulation	3.12 (1.31)	3.40 (1.10)	−1.39	0.17	−0.25
Trusting	3.56 (1.26)	3.73 (0.97)	−0.74	0.46	−0.13
Psychological symptoms	21.14 (15.63)	19.59 (15.55)	0.84	0.41	0.16
**Disordered eating symptoms**	**1.22 (0.62)**	**0.96 (0.62)**	**2.68**	**0.01**	**0.48**
Suicidal ideation	1.00 (2.02)	0.71 (1.74)	1.72	0.10	0.31

*Note.* Statistically significant differences appear in bold text.

## Data Availability

Data are available from the first author upon request.

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
