# Peer review of "A Pilot Feasibility Study of Reconnecting to Internal Sensations and Experiences (RISE), a Mindfulness-Informed Intervention to Reduce Interoceptive Dysfunction and Suicidal Ideation, among University Students in India"

_brainsci, 2022, doi:10.3390/brainsci12020237_

Round 1
Reviewer 1 Report
Suicide is a major public health problem globally. This study aimed to evaluate the relationship between interoceptive dysfunction and suicidality. It also evaluated the feasibility and acceptability of the RISE intervention, a mindfulness intervention that targets interoceptive dysfunction. While the manuscript describes promising data, there are some issues to address:
Comments:
- General comment: If data categorized as “western” is solely from the USA, please use “US” as it is not necessarily representative of data from other countries that would be similarly categorized as “western”.
Introduction:
- Page 2 Ln 47: If the west is referring to the US, then please change “west” to “US”. Additionally, the data referenced is very old. In the US, the highest suicide rates in the most recent data (2010-2019) indicate suicide rates are highest among those aged 45-64 (https://wisqars.cdc.gov/cgi-bin/broker.exe). Also, the rates of suicide are increasing most among younger populations in the US.
- Page 2 Ln 49: Please provide the age range defining “young adults”.
- Page 2 Ln 85: Please tone down the statement about DBT. It is a promising treatment for suicidality, but more evidence is needed to show it is highly effective.
- Page 3 Ln100: Please add data about the proportion of people in India with access to the internet and devices if known. If this number is very low, then the intervention’s reach will be seriously limited. If access is possible only among certain subsets of the population, please indicate which ones and the proportion of the total population they represent.
Materials and Methods:
- Page 3 Ln 183: Please describe how correlations were calculated.
- Page 4: Please indicate how effect sizes were calculated.
Results:
- Table 1: While correlations showed p values <.05, the r values (assuming they are Spearman r values) indicate weak correlation, which likely mean small effect sizes. The only correlations that are not weak appear to be among emotion awareness, self-regulation, body listening, body trust for the MAIA subscales and among the current, past and future SI. Not distracting and not worrying seem to have weak correlations with SI; but the medium effect size for not distracting seems overly large.
- Page 6 Ln 225: Another limitation is that while suicidal ideation is important to measure, it is not nearly as predictive of future suicidal behaviors as are past behaviors and number of past behaviors. Further, with the scale used, it is not possible to differentiate between passive SI and SI with intent and/or planning, which raise the risk of suicide substantially (Innov Clin Neurosci.2014 Sep-Oct; 11(9-10): 23–31). Therefore, the correlation found between the two ID subscales and SI may have limited use for preventing suicide.
- Page 6 Ln 232: The interoceptive dysfunction subscales showed no or weak correlation with suicidal ideation.
- Page 7 Ln294: Please include the 3 questions in the TAQ and mention whether TAQ is a validated scale or a scale designed for this study.
- Page 8 Ln329: When were the pre and post-test surveys administered and how were they administered?
- Page 8 Ln335-340: Please provide a specific definition for feasibility. The consent and participation rates should be mentioned (40/50= 75% and 31/40=78%). The attrition rate was quite high given participants were paid.
- Page 8 Ln341-344: All acceptability data should be presented, preferably in a table.
- Page 8 Ln325: Please calculate FDR to adjust for multiple comparisons and possible Type I errors. Include these adjusted data in Table 2.
- Page 8 Ln345: Please change Primary Clinical Outcomes to Secondary Outcomes. The primary outcome was acceptability, and the study was not powered to measure the other outcomes.
- Page 8 Ln 346: There was no significant change in a majority of outcomes. Please remove any mention of any results that were not significant or trending.
- Table 2: The effect sizes listed in the table do not match that in the text.
- Page 9 Ln369: A major limitation that was not mentioned is the subscales affected by RISE were not those found with weak association to SI, though the Not worry subscale might be trending. Therefore, there is no evidence that RISE will decrease SI in the population studied especially when part of a controlled clinical trial.
General Discussion:
- Page 9 Ln403: The acceptability data was 6.3 in the Results section, not 7 as listed here.
- Page 9 Ln405: Show evidence that a 22% attrition rate is good compared to similar interventions.
- Page 9 Ln408: All aspects of interoceptive dysfunction did NOT improve significantly. Please only list the significant improvements as any insignificant findings are likely to be lost in an RCT.
Reviewer 2 Report
This manuscript is entitled “A pilot feasibility study of a mindfulness informed intervention to reduce interoceptive dysfunction and suicidal ideation among university students in India”. This study comprises two studies. Study 1 examined whether the relationship between interoceptive deficits and suicidality observed in western populations is replicated in an Indian population, as a preliminary study for the intervention study. Study 2 examined the effectiveness of a mindfulness-informed intervention aimed at improving interoception (RISE) on suicidality. This study is providing valuable data demonstrating the relationship between interoception and suicidality in India as well as evaluating a novel program for the improvement of interoception.
Major issues
<Study 1>
・Please clarify how the authors determined the study sample size.
・Line 120, in study 1, “Participants were included if they were above 18 years of age, were fluent in English, of Indian origin, and living in India.” In my understanding, all materials utilized in the study are provided in the English language. How did the authors confirm that they are fluent in English? Based on their self-report? This may be an important issue as it affects the validity of measures and the effectiveness of the program.
・Line 129, in study 1, ”To ensure data quality, three attention checks were placed throughout the online survey. Participants who failed two or more attention checks were screened out of the study, leaving us with a final analytic sample of 276 individuals.” Of all participants initially recruited (n = 465), only 59.4 % of participants were included in the final analysis. This may be problematic. In my speculation, the relatively long time needed to complete the survey (56 min on average) may have affected their concentration. Particularly, it is possible that individuals with depressive symptoms or suicidality tend to show impairment in cognitive functioning, and thus the attention checks may have rather caused biases in the result. This issue regarding attrition by attention checks should be noted in the limitation section. At the same time, the authors should provide more details about “attention checks”. This information would be useful for future studies to plan a better procedure for data quality assurance.
・The authors state that the hypothesis about the relationship between interoception and suicidality was partially confirmed, but some correlations are rather weak and almost negligible (rs < .20), although significant. The authors should be more conservative when interpreting these correlations.
<Study 2>
・Please clarify how the authors determined the study sample size.
・The authors should report the breakdown of the attrition (i.e. dropouts or failures in attention checks). As in study 1, I am wondering about how many participants were removed due to the attention checks.
Minor issues
・I recommend the authors include the name of the program (RISE) in the manuscript title, abstract, or Keywords (but not necessary).
